# MAKE OPTIMIZATION ONCE AND FOR ALL WITH FINE-GRAINED GUIDANCE

## ABSTRACT

Learning to Optimize (L2O) enhances optimization efficiency with integrated neural networks. L2O paradigms achieve great outcomes, *e.g.*, refitting optimizer, generating unseen solutions iteratively or directly. However, conventional L2O methods require intricate design and rely on real optimization processes and numerical optimization results, limiting scalability and generalization. Our analyses explore general framework for learning optimization, called *Diff-L2O*, focusing on augmenting sampled solutions from a wider view rather than local updates in real optimization process only. Meanwhile, we give the related generalization bound, showing that the sample diversity of Diff-L2O brings better performance. This bound can be simply applied to other fields, discussing diversity, mean-variance, and different tasks. Diff-L2O's strong compatibility is empirically verified with only minute-level training, comparing with other hour-levels.

## 1 INTRODUCTION

Learning to optimize (L2O) (Chen et al., 2017; 2022b; Metz et al., 2022; Li & Malik, 2016) aims to improve the efficiency of optimization algorithms by refitting optimization algorithms with (machine) learning. Learning optimization algorithms involved in iteration, it has significant advantages in accelerating optimization algorithms (Chen et al., 2022a; Xie et al., 2024; Zheng et al., 2022; Cao et al., 2019).

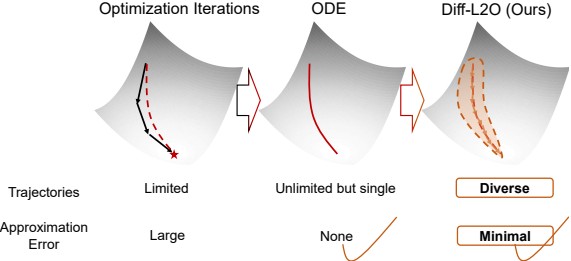

Figure 1: Diff-L2O's intuitions: wider views and better sampling diversity on solution spaces.

Popular L2O algorithms with great performance are usually composed of the following paradigms. 1) Learning the settings of the optimizer so as to (Xie et al., 2024) find a set of settings that make the optimizer search the solution space faster and more stable; 2) using a generator to guide the model iteration, e.g., iterating the model step by step with the inference of a sequence model (Chen et al., 2017); 3) modeling the parameter space directly and generating the parameters of the model in a better way (Gartner et al., 2023).

However, L2O methods require delicate design and tuning, depending on real optimization processes. These paradigms 1) do not directly model the optimization process in general but each point on trajectories or 2) rely on the real optimization process of specific types of optimizers. These facts limit L2O scaling up (Metz et al., 2022), and loss the advantage of the generalization capabilities brought by machine learning. Ours below helps solving potentially unknown optimization problems w/o sophisticated designs.

Corresponding to the two aforementioned points respectively, discussion is about 1) the feasibility of unified modeling (Attouch et al., 2019) for the vast majority of optimization algorithms, and the corresponding optimizers, by means of unified modeling (Xie et al., 2024); 2) propose a optimization with wider views, *i.e.*, find a range to the solution, rather than finding a locally best update direction. We explore the generalization performance under this unified modeling and give the generalization bound. We brief the main analyses that *augmentation with diffusion improves generalization* of the modeled solutions.

Empirically, the proposed Diff-L2O demonstrates adaptability to quickly obtain initial points and further speed-up optimization for classic optimizers. Only second-level training time cost are needed for Diff-L2O, comparing with other hour-level methods. It also works on deep neural networks.[1] The contributions of this work are as follows:

- We propose a fast method for solving optimization problems using diffusion models while combining artificial and real data with guidance information.
- We analyze the key factors that can be used to model the solution space with generative models, as well as general formulation, and related generalization bound.
- Experiments using diffusion models to model the solution space, thus accelerating optimization, have yielded impressive results with the proposed Diff-L2O.

## 2 METHODOLOGY

### 2.1 PRELIMINARY

**Optimization's general trajectory formulation.** The dynamics of optimization methods, Inertial System of Hessian-driven Damping (Attouch et al., 2019) (ISHD), can be represented as:

$$\ddot{x} + \frac{\alpha}{t}\dot{x} + \beta\nabla^2 f(x)\dot{x} + \gamma\nabla f(x) = 0, \tag{1}$$

where $\nabla$ and $\nabla^2$ are the gradient and Hessian operations respectively, $\dot{x}$ and $\ddot{x}$ are the first and second ordered derivatives of $x$ on time $t$, and, $\alpha$, $\beta$ and $\gamma$ are hyperparameters on $t$ (which abbreviates $\alpha_t, \beta_t, \gamma_t$) that determine the trajectories of the optimization algorithms.

In L2O cases, we want to learn the solution space of the problem $\min_x f(x)$. The model is actually approximating the ODE (*i.e.*, the $\alpha$, $\beta$ and $\gamma$).

**Discretization.** Euler discretization is an efficient and commonly used discretization method. It is primarily affected by non-linear sampling scenarios. In such cases, the rugged and unknown real optimization surface limits the possibility of further acceleration Xie et al. (2024); Schuetz et al. (2022) and can easily lead unstable results.

**Stochastic optimization's dynamics.** The dynamics in Equ. 1 is the general ODE of the most gradient-based optimization trajectories. However, more practical dynamics are stochastic ones, which can be represented by stochastic differential equations (SDE, Ito formula of Wiener process) $\mathbf{d}\tilde{x} = u\mathbf{d}t + v\mathbf{d}w$, where $w$ is the Brownian motion, $u$ and $v$ are the functions on $t$ determining the types, which abbreviates $u_t$ and $v_t$.

**Diffusion process.** The aforementioned classic formulation of a diffusion process is not enough since due to direct expression of different common stochastic processes. So we have the following more specific ones. In a more general case, we reformulate it into the following one.

$$\mathbf{d}\tilde{x} = \tilde{x}\dot{s}/s\mathbf{d}t + s\sqrt{\dot{\sigma}\sigma}\mathbf{d}w, \ \tilde{x} = s\tilde{x}_0 + s\sigma\epsilon, \epsilon \sim \mathcal{N}(0, \mathbf{I}), \tag{2}$$

where $\sigma$ and $s$ abbreviates $\sigma_t$ and $s_t$, $\tilde{x}_t$ is the stochastic process with given $\tilde{x}_0$ as initial point.

### 2.2 DISCUSSION: MODELING SOLUTIONS IS FEASIBLE

We give an intuitive discussion in this section. See Sec. 2.3 for more details.

**Takeaways.** Our discussion is summarized below.
1) Optimization process's meta features do provide information for solution space modeling;
2) The data from the real optimization process is helpful, but it is still not enough.

**Case: overparameterization.** We know that optimization algorithms have their own implicit biases (or regularization) (Gunasekar et al., 2018a), when the case goes with overparameterization, *e.g.* small norms, sparse solutions, flat (stable) solutions, small gradients, and maximum margin.

The implicit biases (Dauber et al., 2020; Soudry et al., 2018; Gunasekar et al., 2018b) depend on the problem formulation and the optimization algorithm. which means that the *optimization formulation*

---

[1]Results on DNN are in the Appendix.

*and algorithm is informative* to the expected results. Linear regression, for example, tends to a min-norm solution with the gradient descent optimizer.

**Case: underparameterization.** The implicit biases within under-parameter classical problems (Bowman & Montúfar, 2022) can be reduced into subspaces. For example, linear regression can be full-ranked on subspaces, maintaining the similar solution spaces with the form of implicit bias.

**Case: low performance.** Moreover, low performance in the under-parameterized case would not be directly related to the feasibility of solution spaces being modeled. It would make the surface more mundane and some SDEs more chaotic. Performance is low, yet the parameter space is easy to approximate, because the prediction only needs to be noise, given the targeted chaotic SDE.

Thus, the optimizer, the optimizee (*i.e.* problem itself), and other meta-features are all informative.

**Closest doesn't mean best.** Different implicit biases imply different probability distributions of solutions. Unexplored implicit biases could bring better solutions within the solution space. The closest approximations to the trained solutions or the converged SDEs are thus not the best. Decoupling dependency on real optimization trajectories is a greater potential for generalization.

**The closest is yet informative.** Well-fit-SDE models can still tell us a lot. For example, in the case where *mode connectivity* (Garipov et al., 2018) is considered, the terminal phases of the optimization SDEs do not exactly converge, but rather swim around within a connected region toward the similar-performance region that meets the implicit bias.

We conclude that effective parameter space modeling is *diverse and trajectory-guided*.

### 2.3 DIFF-L2O: HOW TO MODEL SOLUTIONS

According to the discussion, our approach focuses on using 1) trajectories from the optimization process as guidance, and 2) both real and artificial SDE to ensure validity and exploration.

**Artificial trajectories: diffusion process.** Random noise is introduced to explore more potential solutions near optimization trajectories. These potential solutions should follow real SDE to make full use of the real optimization. These trajectories start from suboptimal solutions, with smooth connections between them, thereby exploring potential solutions in the surrounding area.

The diffusion process is simulated according to the current big-hit diffusion models. The diffusion processes' general forms are shown in Equ. 2 and specialized in Tab. 1, including DDPM (VP-SDE) (Ho et al., 2020), VE-SDE (Song et al., 2021) and EDM(Karras et al., 2022).

Table 1: The ingredients of SDEs.

| SDEs | VP | VE | EDM |
|---|---|---|---|
| $s$ | $\exp\{-\frac{1}{4}\Delta_\beta t^2 - \frac{1}{2}\beta_0 t\}$ | $1$ | $1$ |
| $\sigma^2$ | $\exp\{\frac{1}{2}\Delta_\beta t^2 + \beta_0 t\} - 1$ | $t$ | $t^2$ |
| $\dot{s}$ | $-\frac{\sigma\dot{\sigma}}{(1+\sigma^2)^{3/2}}$ | $0$ | $0$ |
| $\dot{\sigma}$ | $\frac{(1+\sigma^2)(\Delta_\beta t+\beta_0)}{2\sigma}$ | $1$ | $2t$ |

$\triangleright \beta_0$ and $\Delta_\beta$ are pre-defined parameters.

**Discretization and sampling.** We use the simple and efficient Euler sampler. The SDE is isotropic diffusion using DDPM (VP-SDE) (Ho et al., 2020; Song et al., 2021). The sampling algorithm are shown in Algorithm 1 and Algorithm 2.

---

**Algorithm 1** Forward Scheduling

**Inputs:** The starting point of the forward trajectory $\tilde{x}_0$, and a coefficient list $[\bar{\alpha}_0, \ldots, \bar{\alpha}_T]$

    **for** $t = 1, 2, \ldots, T$ **do**
        $\tilde{x}_t \leftarrow \mathcal{N}(\sqrt{\bar{\alpha}_t}\tilde{x}_0, (1-\bar{\alpha}_t)\mathbf{I})$
    **end for**
**Output:** $[\boldsymbol{x}_0, \boldsymbol{x}_1, \ldots, \boldsymbol{x}_T]$

---

**Algorithm 2** Backward Sampling

**Inputs:** A standard Gaussian noise $\hat{\boldsymbol{x}}_T \sim \mathcal{N}(0, \mathbf{I})$, and a guidance vector $\boldsymbol{g}$.

    **for** $t = T, T-1, \ldots, 1$ **do**
        $\boldsymbol{t} \leftarrow \text{TE}(t)$
        $\hat{\boldsymbol{x}}_{t-1} \leftarrow \text{opt}(\text{concat}(\hat{\boldsymbol{x}}_t, \boldsymbol{g}, \boldsymbol{t}))$
    **end for**
**Output:** $\hat{\boldsymbol{x}}_0$

---

**Training: Diff-L2O.** Since our approach is Euler sampling on VP-SDE, we use $\epsilon$-parameterization to train our diffusion model, according to DDPM. However, DDPM does not consider how the solution behaves in the optimization process, only whether it is aligned well with white noise.

Our approach uses the aforementioned guidance (*e.g.*, quantities in the processes, optimization meta-features). These help the parameter space modeled to be embedded with meta-information

about optimization. This brings greater generalizability. Meanwhile, we add the loss of the current solutions on the optimization objective as a metric that is integrated uniformly into the probabilistic modeling of the generated model (Algorihtm 3).

**Generalization analyses.** Diff-L2O augments the diversity of the samples and hence works better. The relevant theorem on our setting is from the perspective of PAC-Bayesian.

The generalization gap is defined as: $\Delta(\hat{x}) := \Delta(\hat{f}_S, \hat{f}_D)$, where $\hat{f}.$ abbr. $f(\hat{x}; \cdot) := \mathbf{E}_{d\sim}.f(\hat{x}; d)\}$. $\hat{f}.$ and $f.$ are the problems' expectation values of $\hat{x}$ and $x$ on probability from approximated model $q$ or the real solution space distribution (*w.r.t.*, min for simplification), $D$ and $S$ are the population (test) and samples (train), *i.e.*, ground truth and sampled solutions in L2O. $\Delta$ abbr. distance $\Delta(\hat{x})$.

---

**Algorithm 3** Diff-L2O Training

**Inputs:** Initial point $\hat{x}_\mathtt{T} \sim \mathcal{N}(0, \mathbf{I})$, guidance vector $g$, the optimizee's parameter $\theta$, the forward trajectory $\{\tilde{x}_0, \tilde{x}_1, \ldots, \tilde{x}_\mathtt{T}\}$, loss coefficient $\alpha$

    **for** $t = \mathtt{T}, \mathtt{T} - 1, \ldots, 1$ **do**
        $\boldsymbol{t} \leftarrow \mathtt{TE}(t)$
        $\hat{x}_{t-1} \leftarrow \mathtt{opt}(\mathtt{concat}(\hat{x}_t, g, \boldsymbol{t}))$
        $\mathcal{L}_1 \leftarrow f(\theta, \hat{x}_{t-1})$
        $\mathcal{L}_2 \leftarrow \mathrm{MSE}(\tilde{x}_{t-1}, \hat{x}_{t-1})$
        $\mathcal{L} \leftarrow \alpha\mathcal{L}_1 + (1 - \alpha)\mathcal{L}_2$
        Update $\mathtt{opt}$ by minimizing $\mathcal{L}$
    **end for**

---

This differs the previous PAC-Bayesian bounds in the artificial samples' distribution and $\hat{x}_t \sim q_t(g)$ obtained from a stochastic process of guidance $g$, *e.g.*, meta-features. The time $t$ and condition $g$ are omitted for simplicity below.

**Theorem 2.1.** *(General PAC-Bayesian on stochastic solution space.) In this general theorem, $\Delta$ requires only a non-negative general convex distance, and we do not restrict the optimization objective to the downstream tasks. With a initial prior process $p$, $\forall q$ (posterior) w/ $n$ #samples, we have the following bound at least $1 - \delta$ probability:*

$$\Delta \leq_{1-\delta} \frac{1}{n}\{\mathrm{KL}(q||p) + \log \frac{\mathcal{M}}{\delta}\}, \forall time\ t$$

*where $\mathcal{M} := \mathbf{E}_{h\sim p} \exp\{n\Delta(h)\}$ is related to the optimization task, including the distance between population and the training set.*

*Proof.* With given probability $1 - \delta$ (w.h.p.), we have $\Delta(\hat{f}_S, \hat{f}_D) \leq \epsilon_\delta(n)$. As our problem is defined as min for simplification, we focus on the upper bound here.

From the expectation extended objectives: $\hat{f}_D = \mathbf{E}_{\hat{x}\sim q}\Delta$ and $\hat{f}_S = \mathbf{E}_{\hat{x}\sim q}f(\hat{x}; S)$, we decouple a prior $p$ from modeled distribution $q$ with Jensen inequality, $\log \mathbf{E}_{h\sim p} \exp\{n\Delta(h)\} \geq n\Delta - \mathrm{KL}(q||p)$. With Markov inequality, introducing probability 1-$\delta$, $\Delta \leq \frac{1}{n}\{\mathrm{KL}(q||p) + \log \frac{\mathcal{M}}{\delta}\}$, w.h.p., where $\mathcal{M} := \mathbf{E}_{h\sim p} \exp\{n\Delta(h)\}$ is independent of $q$. It should be discussed in different optimization objectives and downstream tasks. The all do not depend on time $t$ here. $\square$

General generalization upper bounds are time-independent, and next we discuss specific SDE modeling processes that are time-dependent, and their relationship to tasks.

**Corollary 2.2.** *(Diff-L2O: Gaussian.) When $p \sim \mathcal{N}(\mu, \Sigma)$, $q \sim \mathcal{N}(\hat{\mu}, \hat{\Sigma})$, the KL-divergence is*

$$\mathrm{KL}(q||p) := \frac{1}{2}\{\log \frac{|\Sigma|}{|\hat{\Sigma}|} - k + ||\hat{\mu} - \mu||_\Sigma^2 + \mathrm{tr}(\Sigma^{-1}\hat{\Sigma})\}.$$

*In Diff-L2O, the Gaussian is isotropic, and initial prior $p \sim \mathcal{N}(\sqrt{\bar{\alpha}_t}x, (1 - \bar{\alpha}_t)\mathbf{I})$, $x \sim D$. We can further format the bound as*

$$\Delta \leq_{1-\delta} \frac{1}{n}\{k \log(1 - \bar{\alpha}_t) - \log|\hat{\Sigma}| - k + ||\hat{\mu} - \mu||_2^2 + \frac{\mathrm{tr}(\hat{\Sigma})}{(1 - \bar{\alpha})} + \log \frac{\mathcal{M}}{\delta}\}, where\ k = \dim x.$$

**Corollary 2.3.** *(Diff-L2O: Classification tasks.) Generalizing over the classification task, we define $\hat{f}_D$ and $\hat{f}_S$ by considering the prediction error rate of the modeling probability $q$ on the test and training sets, and use the difference between the two as the distance $\Delta$.*

*If the error rate is $m/n$ ($m$ misclassified samples among $n$ samples), we have the probability:*

$$\mathbf{P}_{\tilde{S}\sim D}(\hat{f}_S = m/n) = \mathrm{Bio}(m; n, \hat{f}_D), \forall m,$$

*where $\tilde{S}$ is a set of $m$ independent samples. We have:*

$$\mathcal{M} = \sup_{\mathcal{P} \in [0,1]} [\sum_{m=0}^{n} \text{Bio}(m; n, \mathcal{P}) \exp\{n\Delta(m/n, \mathcal{P})\}]$$

Thus, we have the following bound, when Diff-L2O is applied to general classification tasks or other tasks that can be reduced into classification.

$$\Delta \leq_{1-\delta} \underbrace{\frac{k}{n}[\log(1 - \bar{\alpha}_t) - 1]}_{\text{diversity} \uparrow} + \underbrace{\frac{||\hat{\mu} - \mu||_2^2}{n}}_{\text{about bias} \downarrow} - \underbrace{\frac{\log |\hat{\Sigma}|}{n} + \frac{\text{tr}(\hat{\Sigma})}{n(1 - \bar{\alpha})}}_{\text{about variance} \downarrow}$$

$$+ \underbrace{\log \frac{1}{\delta} \left( \sup_{\mathcal{P} \in [0,1]} [\sum_{m=0}^{n} \text{Bio}(m; n, \mathcal{P}) \exp\{n\Delta(m/n, \mathcal{P})\}]) \right)}_{\text{about task}(\textit{i.e.}, \text{ the optimizee})} .$$

**Takeaways.** From the bound, we know that:

- For any stochastic process at any time $t$, is a Gaussian distribution, the solution's dimension $k$ have to *grow linearly* with the sample size $n$.
- A larger sample size $n$ reduces the generalization gap, *i.e.*, sum of bias and variance. At a certain overall loss (*e.g.*, the terminal phase of training), there is a classical bias-variance trade-off.
- The ability to generalize is also related to the kind of downstream task, with specific effects $\mathcal{M}$. As in the above example, $\mathcal{M}$ usually takes supremum for further concentration.

**Theorem expansion.** Here we use the general distribution assumption for the stochastic process. Markov inequality in the proof can be replaced with different assumptions, *e.g.*, using Hoeffding inequality for the sub-Gaussian, Bernstein inequality for the sub-exponential.

**Theorem specialization.** Given different assumptions and tasks *w.r.t.* $\mathcal{M}$ and $\Delta$, we have the Table 2. Previous works are related in order (Langford & Seeger, 2001; McAllester, 1998; Alquier & Guedj, 2018).

| bound modifications *w.r.t.* $\Delta(a, b)$ on the left-hand side |
|---|
| $a \log \frac{a}{b} + (1 - a) \log \frac{1-a}{1-b} \leq \frac{1}{n}[\text{KL}(q\|p) + + \log \frac{\sqrt{2n}}{\delta}]$ |
| $(b - a)^2 \leq \frac{1}{2n}[\text{KL}(q\|p) + \log \frac{\sqrt{2n}}{\delta}]$ |
| $b - a \leq \frac{1}{\lambda}[\text{KL}(q\|p) - \log(\delta) + \frac{\lambda}{n}(b - a)]$ |

Table 2: Specialization: varied distance function $\Delta$.

### 2.4 ADD-ON: OPTIMAL-FREE AND DIMENSION-FREE

`oracle` is a neural network to generate initial points. It learns from the suboptimal solutions, and training from scratch is avoided. An element-wise variant for dynamic dimension $k = \dim x$ is provided in the Appendix.

---

**Algorithm 4** Alternative `oracle`: optimal generator

**for** given #epochs **do**
    $x_0 \leftarrow \texttt{oracle}(g)$
    $\mathcal{L}_{\text{pre}} \leftarrow f(\theta, x_0)$
    Update `oracle` by minimizing $\mathcal{L}_{\text{pre}}$
    Generate the forward trajectory starting from $x_0$: $\{\tilde{x}_0, \tilde{x}_1, \ldots, \tilde{x}_{\text{T}}\}$
    Train `opt` using Algorithm 3 for one epoch
    $\mathcal{L}_{\text{post}} = \text{MSE}(x_0, \hat{x}_0)$

    Update `oracle` by minimizing $\mathcal{L}_{\text{post}}$
**end for**

---

## 3 EMPIRICAL EVALUATION

### 3.1 OVERVIEW

Numerical evaluations are built on conventional optimization problems, including convex and non-convex cases. Diff-L2O is applicable on the parameter solution space of the neural network. Summary: 1) Diff-L2O improves the conventional optimizers well; 2) vanilla Diff-L2O also works well on non-convex problems.

### 3.2 SETTINGS

**Compared baselines.** We compare various analytical optimizers (Gradient Descent and Adam (Kingma & Ba, 2014)) and learned optimizers (L2O-DM (Andrychowicz et al., 2016) and L2O-RNNProp (Lv et al., 2017)). For learned optimizers, we train them on the same set of samples.

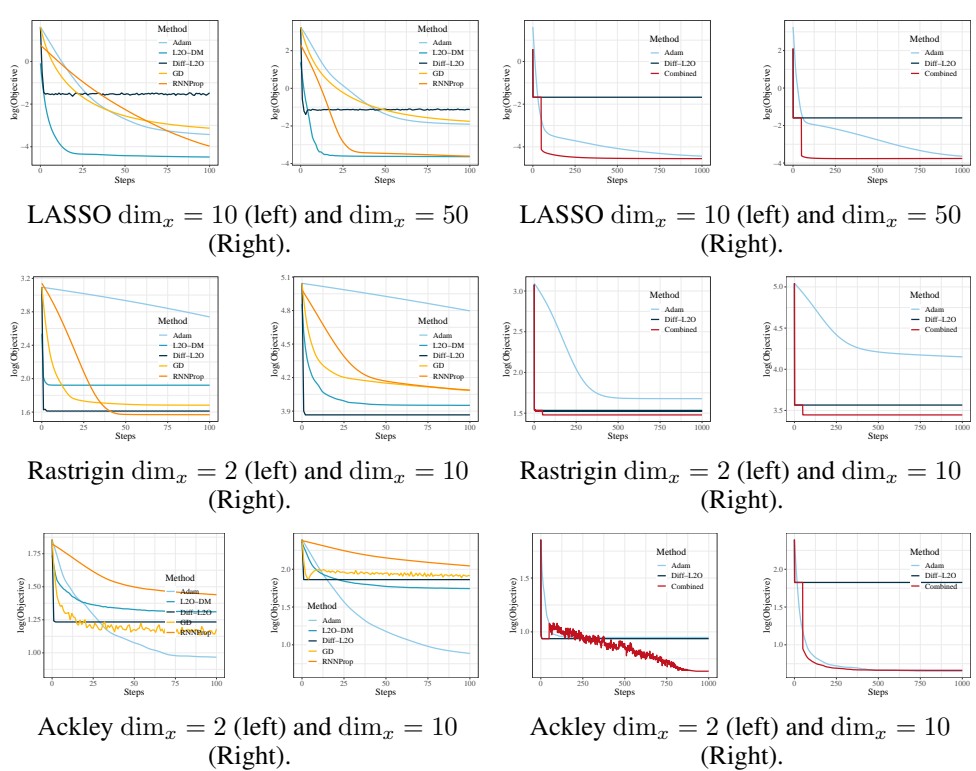

LASSO $\dim_x = 10$ (left) and $\dim_x = 50$ (Right).

LASSO $\dim_x = 10$ (left) and $\dim_x = 50$ (Right).

Rastrigin $\dim_x = 2$ (left) and $\dim_x = 10$ (Right).

Rastrigin $\dim_x = 2$ (left) and $\dim_x = 10$ (Right).

Ackley $\dim_x = 2$ (left) and $\dim_x = 10$ (Right).

Ackley $\dim_x = 2$ (left) and $\dim_x = 10$ (Right).

Figure 2: Comparison on optimizees across #dimension: LASSO, Rastrigin and Ackley.

Figure 3: Ablation: compatibility of Diff-L2O with conventional optimizers.

**Training hyperparameters.** The maximum step T is set to 100 when training opt. #Diffusion steps for inference is 100. The coefficient for variance scheduling range from $1 \times 10^{-5}$ to $2 \times 10^{-2}$, linearly increasing along $t$. The coefficient $\gamma$ for loss balancing is set to 0.5 as default.

**Optimizees' hyperparameters.** Diff-L2O is evaluated on three representative optimization problems with varied complexities and characteristics. For all optimizees, training and testing samples are independently drawn from a standard Gaussian distribution $\mathcal{N}(0, \mathbf{I})$. For example, in LASSO, $\mathbf{A}$ and $\mathbf{b}$ are sampled from standard Gaussian, simplified as $\theta$.

$$x^{\text{LASSO}} = \arg\min_{\boldsymbol{x}} \ \frac{1}{2}\|\mathbf{A}\boldsymbol{x} - \mathbf{b}\|_2^2 + \lambda\|\boldsymbol{x}\|_1 \tag{3}$$

Other formulations of classic problems about Rastrigin and Ackley are in Appendix.

▷ *LASSO* Two problem scales is related: a low-dimensional setting with design matrix $\mathbf{A} \in \mathbb{R}^{5 \times 10}$ and a medium-dimensional setting with $\mathbf{A} \in \mathbb{R}^{25 \times 50}$. The $\ell_1$ regularization coefficient is fixed at $\lambda = 0.005$ for both configurations.

▷ *Rastrigin* We investigate both low-dimensional ($d = 2$) and high-dimensional ($d = 10$) scenarios. The amplitude of the modulation term is set to $\alpha = 10$, which controls the intensity of local minima. It's non-convex.

▷ *Ackley* Similar to the Rastrigin function, we examine the optimization performance in both low-dimensional ($d = 2$) and high-dimensional ($d = 10$) spaces. It's non-convex.

### 3.3 COMPARISON

**LASSO.** We first conduct experiments on the LASSO optimizees and compare the performance on unseen optimizee problems. The experimental results are summarized in Figure 2. We can observe that Diff-L2O converge faster compared to other baselines, achieving near-convergence range with less than ten steps. In the absence of gradient information, Diff-L2O converges to the wall of the

LASSO convex valley. This issue can be easily resolved by combining Diff-L2O and analytical optimizers to achieve more accurate solutions.

**Rastrigin.** In Rastrigin tasks, our method has demonstrated faster convergence speed and also similar or higher quality compared to baselines. Specifically, Diff-L2O achieves a loss objective of 44.09 within 10 steps, while the most competitive baseline, *i.e.* RNNProp, can only achieve a loss of 56.68 in 100 steps. Such an advantage is enlarged in higher-dimensional cases of the variables as baselines suffer from the curse of dimensionality, while our method performs consistently for different dimensions.

**Ackley.** On the Ackley tasks, Diff-L2O also out-performs existing baseline methods with clear margins: in 2-dimensional case, Diff-L2O achieves a loss objective of 3.15 within 10 steps, compared to the most competitive baseline, *i.e.* RNNProp, which can only achieve a loss of 4.48 in 10 steps. In 10-dimensional case, Diff-L2O achieves a loss objective of 5.37 within 10 steps, while the most competitive baseline can only achieve a loss of 6.08 in 100 steps. Analytical optimizers such as Adam outperform all L2O methods due to the moderate difficulty of Ackley problems.

**MNIST on DNN.** We evaluate the classification performance of Diff-L2O on MNIST. In Figure 7 (Appendix), and it achieved a loss of 0.228 and accuracy of 92.06% on test set, which outperform RNNProp that achieves a loss of 0.268 and accuracy of 90.28, and L2O-DM with a loss of 0.252 and accuracy of 90.79 on the same test set. Detailed settings are in Appendix.

## 3.4 ABLATION

**Ablation: compatibility with conventional optimizers.** *Diff-L2O works well when adapted to other methods.* The stochastic nature of diffusion models enables rapid initial convergence but may slow in later stages, which is particularly disadvantageous for convex problems. This motivates our hybrid approach: the diffusion model starts for initialization and traditional optimizers follow. Our results show hybrid strategy consistently outperforms others on both convex and non-convex cases.

**Settings.** We evaluate all optimizees on the same test set as the comparison experiments. Our hybrid optimization consists of two phases: an initial exploration phase utilizing our diffusion-based model for the first 50 iterations, followed by a fine-grained fine-tuning phase with the Adam optimizer.

**Analyses.** Fig. 2 and 3 show that, in the comparison experiment' convex case, the performance using a vanilla Diff-L2O can be improved by using a combination of conventional optimizers. Diff-L2O can be used to quickly generate foundational solutions with a small amount of fine-tuning to reach the optimal.

**Ablation: optimal-free.** The training of diffusion models requires solving numerous optimization problems of the same optimizee family, which inherently limits the model's generalizability. The `oracle` component offers a potential solution to this limitation. Therefore, we conduct an ablation study to analyze how different oracle configurations impact the model's performance.

**Settings.** We conduct a series of experiments to understand the effects of introduced components: (1) *Noisy*: we replace `oracle` with a module that generates random noises; (2) *Fixed*: we do not update the `oracle` network; (3) *Partial*, we update the `oracle` network with $\mathcal{L}_{\text{pre}}$ only; and (4) *Perfect*: `oracle` output always the optimal solutions.

Table 3: Log loss with varied oracles.

| variants | LASSO | Rastrigin | Ackley |
|---------|--------|-----------|--------|
| noisy | -1.306 | 1.727 | 1.301 |
| fixed | -1.427 | 1.657 | 1.281 |
| partial | -1.456 | 1.627 | 1.257 |
| perfect | -1.676 | 1.532 | 0.936 |
| Ours | -1.660 | 1.601 | 1.233 |

**Analyses.** According to Tab. 3, In *Noisy* case, we find that random initialization with poor performance. It show us that initialization strategy is necessary, even a fixed pre-trained network. Loss term $\mathcal{L}_{\text{pre}}$, lowering task loss, helps by making better initial points. The benefits, however, are increased gradually comparing to perfect cases. Loss term $\mathcal{L}_{\text{post}}$, closing backward and forward processes, shows the importance of samples with great diversity. All these modules lead DIff-L2O's performance closer to the perfect cases (starting at the optimal).

**Ablation: guidance.** In this part, the guidance vector $g$ can be time step $t$ dependent, and we denote it by $g_t$. In practice, $g_t$ is a crucial component in Diff-L2O. For convex problems like LASSO,

incorporating gradient information in the guidance vector can significantly improve the convergence speed and accuracy. However, in non-convex problems such as Rastrigin, the gradient can potentially be a source of noise that guides the solutions to local minima.

**Settings.** we conducted experiments on LASSO and Rastrigin optimizees using three types of guidance vectors: (1) *Gradient*, where only the gradient is considered as the guidance vector; (2) *Global*, where the optimizees' parameters $\theta$ are used as the guidance vector; and (3) *All*, where the guidance vector consists of both gradient and $\theta$.

Table 4: Log loss with different guidance vector.

| variants | LASSO (t=10) | LASSO (t=100) | Rastrigin (t=10) | Rastrigin (t=100) |
|---|---|---|---|---|
| gradient | -3.161 | -4.011 | 3.064 | 2.738 |
| global | -1.674 | -1.673 | 1.532 | 1.532 |
| all | -3.153 | -3.938 | 1.618 | 1.643 |

**Analyses.** In 1) *convex* cases, as shown in Tbl. 4, the gradient largely guides whether the current point is optimal or not and contains useful information. The gradient-only cases are dominated by the first-order information, and thus got a log loss value of -3.161 and -4.011 from -3.153 and -3.938. 2) The convergence in *non-convex* cases is not strictly determined by the gradient, but gradients at samples are still helpful. The result of 1.618 from $t = 10$ converges quickly compared to 1.643 from $t = 100$, and the sampling has not converged in *gradient* case with a gap of $0.326$.

Table 5: Time costs of L2O-DM and Diff-L2O.

| optimizees | L2O-DM | Diff-L2O |
|---|---|---|
| LASSO (5-dim) | ∼ 4 hours | 203 s |
| LASSO (25-dim) | ∼ 6 hours | 376 s |
| Rastrigin (2-dim) | ∼ 2 hours | 310 s |
| Rastrigin (10-dim) | ∼ 2 hours | 393 s |
| Ackley (2-dim) | ∼ 3 hours | 309 s |
| Ackley (10-dim) | ∼ 3 hours | 543 s |

**Evaluation: training time.** Table 5 demonstrates the training time of L2O-DM (Andrychowicz et al., 2016) and our method. It can be clearly seen that the Diff-L2O can be trained rapidly, using merely 2% of time compared to L2O algorithms. This rapid training makes our model practical.

**Settings.** The experiments's default settings are on GPU 1×NVIDIA-A100 and CPU AMD EPYC 7H12 64-Core. #iterations is 100.

**Visualization: trajectories.** We demonstrate that Diff-L2O rapidly approaches the vicinity of optimal solutions in the early stages, notably within the first iteration. **Settings.** We set the dimension for all optimizees (LASSO, Rstrigin, Ackley) to 2 with other hyperparameters the same.

**Analyses.** Even in the non-convex case, Rastrigin, the learned descent trajectory of the optimizer reaches the area around the global optimum in almost the starting iterations.

**Visualization: modeled distribution. Settings.** The dimension of all optimizees are set to 2 and other hyperparameters keep unchanged. The learned and true distributions mean Diff-L2O in default setting and gradient descent, respectively, with 5000 initial points.

**Analyses.** The learned distribution and the distribution gotten from conventional optimizer are matched generally. The diversity of learned distribution are greater.

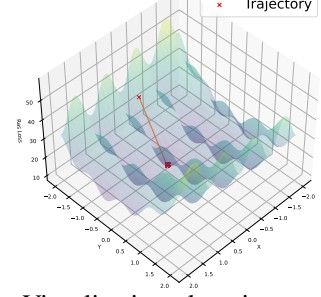

Figure 4: Visualization: learning surface. Fast convergence happens within several epochs.

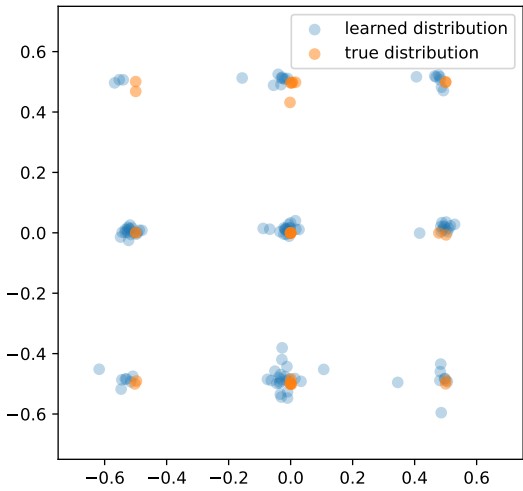

Figure 5: Visualization of the learned and the ground-truth distribution (true). The distributions are generally matched.

## 4  RELATED WORKS

**Learning to optimize (L2O).** L2O is an alternative optimization paradigm that aims to learn effective optimization rules in a data-driven way. It generates optimization rules based on the performance on a set of training problems. it has demonstrated success on a wide range of tasks, including black-box optimization (Chen et al., 2017; Krishnamoorthy et al., 2023), Bayesian optimization (Cao et al., 2019), minimax optimization (Shen et al., 2021; Jiang et al., 2018) and domain adaptation (Chen et al., 2020; Li et al., 2020). More recently, L2O has demonstrated its ability of solving large-scale problems (Metz et al., 2022; Chen et al., 2022b), making it more practical for broader applications, *e.g.*, conditional generation (Wang et al., 2024; 2025; Liang et al., 2025).

The architectures of the learnable optimizer for L2O works have undergone great evaluation. In the seminal work of Andrychowicz et al. (2016), a coordinate-wise long-short-term memory (LSTM) network Hochreiter & Schmidhuber (1997) is adopted as the backbone, which can capture the inter-parameter dependencies with low computational overhead. Subsequently, while some works (Vicol et al., 2021) have utilized multi-layer perceptions (MLPs) for learnable optimization, a large portion of L2O works have adopted the recurrent neural networks (RNNs) Rumelhart et al. (1986) as the architecture of their learnable optimizer (Chen et al., 2021). For example, Shen et al. (2021) proposes to use two LSTM networks to solve min-max optimization problems. Cao et al. (2019) deploys multiple LSTM networks to tackle population-based problems. Later on, researchers have explored the possibility of using Transformers (Vaswani et al., 2017) as learnable optimizers. Chen et al. (2022c) proposes to use Transformer as a tool for hyperparameter optimization. Jain et al. (2023); Gärtner et al. (2023) propose L2O frameworks that apply Transformers to solve general optimization problem and achieves faster convergence compared to traditional algorithms such as SGD and Adam (Kingma & Ba, 2014). In this paper, we propose to apply a different paradigm, *i.e.,* diffusion, as the foundation of our L2O framework. This framework model solution space with a fine-grained approximation.

**Diffusion models.** Diffusion probabilistic models Ho et al. (2020); Song et al. (2020) have emerged as a powerful tool for generating high-quality samples with different modalities such as images (Dhariwal & Nichol, 2021; Rombach et al., 2022), texts Gong et al. (2022); Xu et al. (2023), 3d objects Erkoç et al. (2023); Gu et al. (2023), and videos (Ho et al., 2022). These models have demonstrated on-par or better generation quality compared to their precursors such as generative adversarial networks (GANs) (Goodfellow et al., 2020; Odena et al., 2017; Gong et al., 2019). In a typical training pipeline, diffusion models learn their parameters through iterative addition and removal of noises; and in the inference stage, they begin with a randomly sampled noise and generate the corresponding sample by iteratively denoising. Conditional diffusion models as an important branch of diffusion models, such as those in Ho & Salimans (2022); Liu et al. (2022); Chao et al. (2022), enables generations with clear instruction. In this study, we introduce a novel conditional diffusion model that operates within the solution space of optimization problems including weight of neural networks. Empirically, diffusion models work well.

## 5  CONCLUSION

This work proposes a novel L2O framework Diff-L2O. It uses diffusion model to learn from the solution space, accelerating the optimization process. Diff-L2O achieves great performance by capturing a wider range near the real trajectories, which is supported by theoretical results. We discuss the key to modeling the solution space while giving relevant generalization bound. Diff-L2O is empirically verified to achieve significant results on multiple benchmarks, which further validates our analyses and discussion.

Furthermore, the ablation study reveals the essence of designed components in Diff-L2O, and the combined method demonstrates huge potential for implementing our method as initialization in practice, which is especially useful when analytical properties are essential (*e.g.*, convex cases).

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

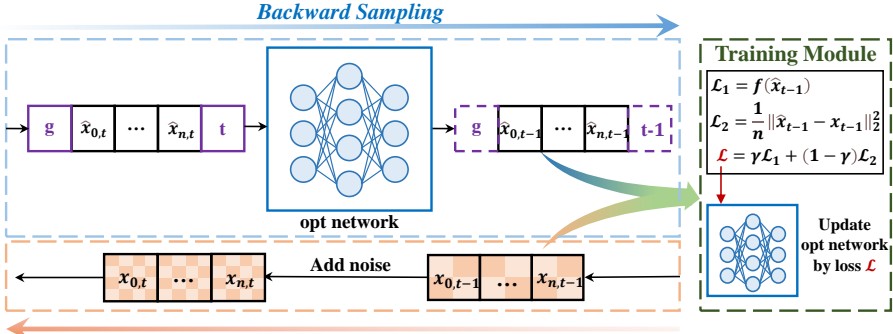

Figure 6: Model Training Framework for Diff − L2O. The lower part is the trajectory generated by forward scheduling before training, and the upper part is the backward sampling from time step $t$ to $t − 1$. Specifically, $\hat{x}_t$ concatenated with guidance vector and time step embedding vector, is passed to the opt network for one-step denoising. Based on $\hat{x}_{t−1}$ and $x_{t−1}$, we calculate the function loss for updating the opt network. (The $x$ in this figure is the $\tilde{x}$ in the main paper.)

## A  GLOSSARY

| name | notation | comment |
|------|----------|---------|
| solution | $x$ | ground truth solutions |
| trajectory | $\{x_t\}_{t\in[T_{\text{train}}]}$ | ground truth trajectories, trained by optimizers |
| blurred solution | $\tilde{x}_t$ | solutions blurred by Gaussian noise |
| blurred trajectory | $\{\tilde{x}_t\}_{t\in[T_{\text{blur}}]}$ | trajectory blurred by Gaussian noise |
| predicted solution | $\hat{x}_t$ | generated by the backward diffusion process |
| predicted trajectory | $\{\hat{x}_t\}_{t\in T_{\text{pred}}}$ | predicted trajectory of diffusion process |
| $\alpha, \beta, \gamma$ | coefficients: SDE | time dependent, especially $\beta$ and $\gamma$ |
| $\mathbf{d}$ | differentiate operator | conventional operator |
| $\nabla, \nabla^2$ | gradient and Hessian matrix operators | conventional operators |
| $\dot{a}, \ddot{a}$ | first and second order derivation of any $a$ | conventional operators |
| $u, v$ | coefficients: time and Brownian motion | determining Wiener process (first order) |
| $s, \sigma$ | parameters: adjustment and intensity | determining general diffusion process |

Table 6: Notations related in this paper.

## B  DETAILED SETTINGS

### B.1  DEEP NEURAL NETWORK ON MNIST

**Model architectures.** We consider the optimizee of MLPs with single hidden layer of dimension 20 and sigmoid activation function, using the cross-entropy loss on the MNIST dataset.

**Optimizees. Optimizees.** To evaluate our model, we deploy the following families of problems as the optimizees.

▷ *Lasso.* We target to minimize the original LASSO objective function without considering the sparsity of the solution:

$$x^{Lasso} = \arg\min_x \frac{1}{2}\|\mathbf{A}x - \mathbf{b}\|_2^2 + \lambda\|x\|_1 \tag{4}$$

where $\mathbf{A} \in \mathbb{R}^{n \times m}$ represent the characteristic matrix of a lasso problem instance, which is fixed and sampled from an *i.i.d.* standard Gaussian distribution. $\mathbf{b} \in \mathbb{R}^{n \times 1}$ refers to the vector of dependent variables, which is also fixed and sampled from an *i.i.d.* standard Gaussian distribution. $\lambda$ is the regularized hyperparameter set to 0.005 in our experiment.

▷ *Rastrigin.* Rastrigin is a common benchmark of non-convex optimization defined in $n$-dimensional space, where $n$ is the number of variables. It is characterized by a complex landscape of multiple local minima and a global minimum. We consider a family of Rastrigin function, and adopt the following definition from a seminal benchmark paper Chen et al. (2017):

$$\boldsymbol{x}^{Ras} = \arg\min_{\boldsymbol{x}} \ \frac{1}{2}\|\mathbf{A}\boldsymbol{x} - \mathbf{b}\|_2^2 - \alpha\mathbf{c}^{\mathsf{T}}\cos(2\pi\boldsymbol{x}) + \alpha n \tag{5}$$

where $\mathbf{A} \in \mathbb{R}^{n\times n}$, $\mathbf{b} \in \mathbb{R}^{n\times 1}$ and $\mathbf{c} \in \mathbb{R}^{n\times 1}$ are all sampled from an *i.i.d.* standard Gaussian distribution.

▷ *Ackley.* Similar to Rastrigin function, Ackley function has many local minima which are comparably larger then the unique global minimum. Compare to Rastrigin, analytical optimizers can find the global minimum with less effort by enlarge their step-size. The problem is definded as:

$$\boldsymbol{x}^{Ack} = \arg\min_{\boldsymbol{x}} \ 20 + e - 20e^{-0.2\|\mathbf{A}\boldsymbol{x}+\mathbf{b}\|_2} - e^{\frac{1}{n}\mathbf{c}^{\mathsf{T}}cos(2\pi\boldsymbol{x})} \tag{6}$$

where $\mathbf{A} \in \mathbb{R}^{n\times n}$, $\mathbf{b} \in \mathbb{R}^{n\times 1}$ and $\mathbf{c} \in \mathbb{R}^{n\times 1}$ are all sampled from *i.i.d.* standard Gaussian distributions.

**Comparison: Loss Curves.** The loss curves between baselines and Diff-L2O are shown in Fig. 7.

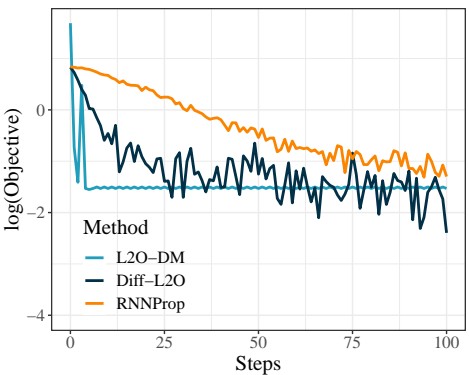

Figure 7: Comparison on MNIST.

## C  THE ELEMENT-WISE VARIANT OF L2O

Algorithm 3 illustrate the Global-to-Local training philosophy by considering three phases, representing early, middle and later phase respectively. For each epoch, we first loop through each time step, and then loop through the positions, i.e. each element of the optimization variable. In early phase, we accumulate the training loss until the last element, called "Global"; In middle phase, we accumulate the training loss and conduct backward propagation on iterating every $\frac{d}{3}$ of elements, which is named Local. In the later phase, where we no longer accumulate the training loss, and this is when element-wise training is achieved.

---

**Algorithm 5** Diff-L2O-ELE Training

---

**Inputs:** $\hat{x}_{\mathrm{T}} \sim \mathcal{N}(0, \mathbf{I})$, a guidance vector $g$, its corresponding trajectory $\{x_0, x_1, \ldots, x_{\mathrm{T}}\}$, phase indicator $\mathrm{N}_1, \mathrm{N}_2$, dimension $d$

    **for** $n = 1, 2, \ldots, \mathrm{N}$ **do**

        **for** $t = \mathrm{T}, \mathrm{T} - 1, \ldots, 1$ **do**

            $t \leftarrow \mathrm{TE}(t)$

            **for** $\mathrm{pos} = 1, 2, \ldots, d$ **do**

                $pos \leftarrow \mathrm{PE}(\mathrm{pos})$

                $x_{t-1,pos} \leftarrow \mathrm{opt}(\mathrm{concat}(x_t, g, t, pos))$

                $\mathcal{L}_1 \leftarrow f(\theta, \hat{x}_{t-1})$

                $\mathcal{L}_2 \leftarrow \mathrm{MSE}(x_{t-1}, \hat{x}_{t-1})$

                $\mathcal{L} \leftarrow L + \gamma \mathcal{L}_1 + (1 - \gamma)\mathcal{L}_2$

                **if** $\mathrm{N} < \mathrm{N}_1$ **then**

                    **if** $\mathrm{pos} == d$ **then**

                        Update $\mathrm{opt}$ by minimizing $\mathcal{L}$

                        $\mathcal{L} \leftarrow 0$

                    **end if**

                **else if** $\mathrm{N}_1 \leq \mathrm{N} \leq \mathrm{N}_2$ **then**

                    **if** $\mathrm{pos} \in \lfloor \frac{d}{3} \rfloor, \lfloor \frac{2d}{3} \rfloor, \lfloor d \rfloor$ **then**

                        Update $\mathrm{opt}$ by minimizing $\mathcal{L}$

                        $\mathcal{L} \leftarrow 0$

                    **end if**

                **else**

                  Update $\mathrm{opt}$ by minimizing $\mathcal{L}$

                  $\mathcal{L} \leftarrow 0$

                **end if**

            **end for**

        **end for**

    **end for**

---

