# OpenReview forum: "Make Optimization Once and for All with Fine-grained Guidance"
_ICLR.cc/2026/Conference — ICLR 2026 Conference Withdrawn Submission_

### Official Review · Reviewer_7QXh · 2025-10-28

**Soundness:** 2
**Presentation:** 3
**Contribution:** 2
**Rating:** 4
**Confidence:** 3

**Summary:**

This paper introduces Diff-L2O, a novel approach to Learning to Optimize (L2O) framed as directly learning the distribution of optimal solutions using a diffusion model, rather than learning an iterative update rule. Conditioned on "fine-grained guidance" (trajectory and objective information), the model aims to generate near-optimal solutions "once and for all." Theoretical analysis via PAC-Bayesian bounds is provided. The method shows rapid meta-training (minutes) and promising results on small-scale classic optimization tasks (LASSO, Rastrigin, Ackley, d<=50) and a tiny MNIST MLP.

**Strengths:**

1. Compared to iterative L2O methods (learning update rules like Adam), using a diffusion model for direct solution generation is a conceptually different and interesting direction.

2. The reported minute-level meta-training is striking compared to typical L2O costs. If scalable, direct generation could significantly cut optimization time.

3. Conditioning the diffusion model on optimization-specific information (trajectories, loss) is a sensible approach to improve generation quality.

4. Providing PAC-Bayesian bounds attempts to ground the approach theoretically, linking diversity to generalization.

**Weaknesses:**

1. The paper positions itself within L2O but largely ignores the highly related field of direct parameter generation using generative models (GANs, VAEs, Diffusion). Is the core idea truly novel compared to prior work using diffusion models to generate network weights directly? The paper fails to compare against or even substantially discuss this relevant literature, some of which may have already demonstrated results on larger scales. This significantly weakens the claim to novelty and impact.

2. The experiments are confined to extremely small-scale, low-dimensional (d<=50) classical problems and a trivial MNIST MLP. This is grossly insufficient to support claims about a general-purpose optimization method for modern deep learning. There is no evidence presented that Diff-L2O can scale to dimensions relevant to DNNs (millions/billions). The claim that it "works on deep neural networks" based only on the tiny MNIST MLP is unconvincing. Can it generate weights for even a ResNet-18 for CIFAR100 task, let alone larger models (refer to the paper *Neural Network Diffusion*)?

3. The ablation study (Fig 3) shows the hybrid method (Diff-L2O init + Adam fine-tune) performs best, contradicting the title's premise. This strongly suggests Diff-L2O is, at best, a potentially good initializer, not a standalone optimizer replacement.

**Questions:**

see weakness

---

### Official Review · Reviewer_jFhw · 2025-11-01

**Soundness:** 2
**Presentation:** 2
**Contribution:** 2
**Rating:** 4
**Confidence:** 3

**Summary:**

This paper introduces Diff-L2O, a new Learning to Optimize (L2O) framework that leverages diffusion training paradigm to accelerate and improve the optimization process. Instead of learning an iterative update rule in prior L2O methods, Diff-L2O models the entire solution space of an optimization problem. The core idea is to treat optimization as a conditional generation task, where a diffusion model learns to denoise a random vector into a high-quality solution conditioned on guidance information (e.g., problem parameters, gradients). Empirically, Diff-L2O is shown to achieve rapid convergence on benchmark tasks. A key claimed advantage is its training efficiency, requiring only minutes of training compared to hours for other L2O methods.

**Strengths:**

- The framework of using diffusion process to model the entire solution space is novel, bringing new insights of learned optimizers.
- The guidance vector $g$ provides a flexible control mechanism, which can encode different requirements of different optimization tasks.
- The experimental results are impressive.

**Weaknesses:**

I think the major weakness is the unclear implementation of the training algorithm (Algorithm 3), include:

- Different training objectives compared to diffusion model. In the manuscript, the authors repeatedly mention the relationship of Diff-L2O and diffusion models. The training objective of diffusion model is to predict the noise hence model $p(x)$. However, Diff-L2O forces the `opt` network to approximate the given trajectory $\tilde{x}$, which is confusing.
- The training process of Algorithm 3 is stateful (i.e., $\hat{x}_0$ depends the operations at timesteps $1,\cdots,T$), which is the most difference compared to standard DDPM. The training paradigm of DDPM is stateless by uniformly sample a timestep $t$ at each training step. Its success is based on complete theory. However, the stateful Diff-L2O training would cause Backpropagation Through Time (BPTT) issue of the gradient flow, which is computationally infeasible, and this paper doesn’t clearly state how Diff-L2O handles this issue (maybe greedy training with truncated BPTT window size of 1?). Besides, the performance could rely on the quality of the trajectory.
- The additional loss $f(\theta, \hat{x}_{t-1})$ is somehow ad-hoc. Would this loss disturb the denoising path and cause training instability?

**Questions:**

- Could you please provide a more detailed explanation of the training loop in Algorithm 3? Specifically, is the `opt` network's loss computed and backpropagated at every reverse step $t$ for a single trajectory? If so, could you elaborate on the motivation for this design choice compared to the standard diffusion model training objective?
- What are the primary computational and memory bottlenecks when applying Diff-L2O to a large parameter vector $x$ (e.g., >1 million dimensions for a standard ResNet)? Can Diff-L2O be used for large-scale deep learning optimization?
- The combined approach (Diff-L2O + Adam) is very effective. Does this suggest that the primary strength of Diff-L2O is as a "warm-start" or initialization method, rather than a full-fledged optimizer capable of finding precise solutions on its own, especially for convex problems?
- Do you ablate on the balance coefficient $\alpha$?

---

### Official Review · Reviewer_P6Yw · 2025-11-01

**Soundness:** 2
**Presentation:** 1
**Contribution:** 2
**Rating:** 2
**Confidence:** 2

**Summary:**

The paper introduces Diff-L2O, a learned optimizer that simulates trajectories using diffusion models. The authors provide theoretical backing for it, and experimental results on small regressions and MNIST.

**Strengths:**

To the extent that I understand the idea, modelling trajectories seems like it could have potential.

**Weaknesses:**

I honestly can't follow the writing throughout (especially when it also needs to contextualize the math), but in specific
I didn't gain much from §2.2: these conclusions/intuitions, to the extent that I understand them, don't seem supported.

The empirical settings are extremely toy, maxing out at MNIST.

a smattering of the grammatical issues:
L81: "easily lead *to* unstable results
L84: "of most gradient-based"

**Questions:**

How expensive is the method? My intuition says running a backward diffusion process would make it extremely so, even if L2O-DM is even worse.

---

### Official Review · Reviewer_zKui · 2025-11-02

**Soundness:** 2
**Presentation:** 2
**Contribution:** 2
**Rating:** 2
**Confidence:** 4

**Summary:**

This paper focuses on the field of Learning to Optimize (L2O) and addresses the key issues of conventional L2O methods, including intricate design, heavy reliance on real optimization processes, and limited scalability/generalization. The authors propose Diff-L2O, which leverages diffusion models to model the solution space, integrates both real and artificial data with fine-grained guidance, and provides a generalizable theoretical bound. The core method of Diff-L2O involves simulating diffusion processes for artificial trajectory generation, using Euler discretization for sampling, and incorporating optimization meta-features as guidance to enhance generalization.
Experiments on synthetic functions and MNIST classifications show the advantages of Diff-L2O.

**Strengths:**

1. The authors conduct in-depth theoretical analysis by deriving a PAC-Bayesian-based generalization bound (Theorem 2.1) and its corollaries for Gaussian distributions and classification tasks.

2. It is acknowledged that the paper’s overall analysis is comprehensive, covering critical validation dimensions: comprehensive ablation studies (e.g., oracle variants in Table 3 , guidance vector effects in Table 4 ) to verify component necessity, training time comparisons (Table 5 ) to demonstrate efficiency advantages, and visualizations (trajectory convergence in Figure 4 , modeled vs. true distribution alignment in Figure 5 ) to intuitively support Diff-L2O’s mechanism.

**Weaknesses:**

1. The paper’s considered experiments are overly simplistic and lack alignment with validation scenarios of recent L2O works. For instance, in the DNN experiment on MNIST, the optimizee is only a simple multi-layer perceptron (MLP) with a single hidden layer of 20 dimensions and sigmoid activation , and the comparison is limited to only two learned optimizers (L2O-DM and L2O-RNNProp) , while recent L2O works often validate on more complex models (e.g., CNNs, Transformers) and diverse datasets (e.g., CIFAR-10, ImageNet) to test generalization.

2. Omission of state-of-the-art L2O methods (as discussed in related work) in all the expeirments.

3. Lack of comparisons with classic analytical optimizers in DNN training. The conventional optimizers (SGD, Momentum-SGD, Adam)—the de facto standard baselines for DNN training—are entirely absent from the MNIST experiment.

4. The general writing should be impoved. The insufficient background hinder the understanding of Diff-L2O’s methodological rationale, especially for readers who are not deeply familiar with L2O or diffusion models. Besides, there are many typos like "Algorihtm 3", "definded".

**Questions:**

See weaknesses.

---

### Note · Authors · 2025-11-19

**Comment:**

Thank you to all reviewers and area chairs for your time and effort. We will further improve the manuscript accordingly.

Best,
Authors of Submission #20959

**Withdrawal Confirmation:**

I have read and agree with the venue's withdrawal policy on behalf of myself and my co-authors.